# Color Reproduction by Multi-Wavelength Bragg Diffraction of White Light

**DOI:** 10.3390/ma16124382

**Published:** 2023-06-14

**Authors:** Alexander Machikhin, Alina Beliaeva, Galina Romanova, Egor Ershov

**Affiliations:** 1Acousto-Optic Spectroscopy Laboratory, Scientific and Technological Center of Unique Instrumentation of the Russian Academy of Sciences, 15 Butlerova, 117342 Moscow, Russia; 2Higher Engineering and Technical School, ITMO University, 49A Kronverksky Pr., 197101 Saint Petersburg, Russia; romanova_g_e@mail.ru; 3Department of Design and Technologies of Electronic and Laser Means, Saint Petersburg State University of Aerospace Instrumentation, 67A Bolshaya Morskaya, 190000 Saint Petersburg, Russia; 4Sector of Color Reproduction and Synthesis, Institute for Information Transmission Problems of the Russian Academy of Sciences (Kharkevich Institute), 19 Bolshoy Karetny, 127051 Moscow, Russia; ershov@iitp.ru

**Keywords:** color reproduction, spectral power distribution, acousto-optic interaction, Bragg diffraction, multi-wavelength light filtration, image processing

## Abstract

Accurate color reproduction is highly important in multiple industrial, biomedical and scientific applications. Versatile and tunable light sources with high color-rendering quality are very much in demand. In this study, we demonstrate the feasibility of multi-wavelength Bragg diffraction of light for this task. Tuning the frequencies and amplitudes of bulk acoustic waves in the birefringent crystal demonstrates high precision in setting the number, wavelengths and intensities of the monochromatic components necessary to reproduce a specific color assigned according to its coordinates in the CIE XYZ 1931 space. We assembled a setup based on multi-bandpass acousto-optic (AO) filtration of white light and verified the reproduced color balance in multiple experiments. The proposed approach delivers almost full coverage of the CIE XYZ 1931 space and facilitates building compact color reproduction systems (CRSs) for various purposes.

## 1. Introduction

Light sources with the ability to accurately display colors play an important role in solving various tasks including physiological studies of visual perception [1], colorimetry [2] and multi-spectral imaging [3]. These devices are a viable alternative to printed color charts and checkers that degrade and change their properties over time. Such tunable illuminants combine light beams from a few sources with different emission spectra, e.g., lasers [4,5] or LEDs [6,7,8,9,10,11,12]. By optical coupling and adjusting the relative intensities of these beams, one can reproduce a specific color with additive color mixing. Because of metamerism, the color may appear the same even when the spectral power distributions (SPDs) of the illuminants differ.

To measure color appearance (from a human observer point of view), the Commission international de l’eclairage (CIE) introduced a special standard CIE XYZ in 1931 [13]. Obviously, it is possible to introduce such a standard because the majority of people with normal color vision have almost the same color perception mechanism. The CIE XYZ 1931 standard introduced the shape of human eye spectral sensitivity functions calculated from the results of color-matching experiments [14] on 17 middle-aged British people with normal color vision. Later similar experiments have been conducted many times to verify the obtained results and investigate the influence of different factors, such as increased field of view for color stimuli [15]. Nowadays, this standard is one of the most important foundations in color reproduction science, because it facilitates the assignment of a numerical color value to the observed spectrum. CIE XYZ 1931 paves the way to separate the brightness (Y) of a color (determined by the total energy of radiation) and the chromaticity by introducing *x*, *y* coordinates. In this paper, we use the CIE XYZ 1931 standard to demonstrate CIE *x*, *y* coverage using different CRSs and estimate the color reproduction quality.

Figure 1 illustrates the areas in this space occupied by typical CRSs based on four lasers [4] and three [11] and four [8] LEDs (Figure 1). We may see that a significant number of colors is not available for reproduction using such devices. To expand their scope, more light sources with different SPDs are necessary, which means a complication of the CRS’s optical design, adjustment and management.

An alternative approach to full-color reproduction might be related to tunable multi-bandpass spectral filtration of white light. Simultaneous selection and mixing of a few monochromatic or narrow-band components of the required wavelengths and intensities could potentially reproduce any color. To implement this approach, one needs a spectral component that is able to transmit light in a few selectable wavelength bands simultaneously with a tunable transmission for each of them. None of the conventional spectral elements (prisms, diffraction gratings, liquid crystal or Fabry–Perot tunable filters, etc.) delivers such a collection of features.

AO interaction, i.e., diffraction of light by ultrasonic waves, is a unique technique that allows tunable multi-bandpass filtration. It became a flexible and repeatable method for selecting the required spectrum of light [16]. That is why we believe that AO interaction may become an effective tool for selecting a specific color from wide-band light and propose applying it for color reproduction tasks. This feasibility study aims to experimentally validate this idea and comprehensively evaluate its advantages and limitations.

AO spectral filtration of light as well as other AO effects are based on photoelasticity, i.e., the change in the refractive index of a medium due to the presence of sound waves that create a volume diffraction grating. By changing the parameters of ultrasound, one may vary the structure of this grating and the diffraction pattern. In Bragg mode, when 2π*λL* >> Λ^2^ (*λ* and Λ are wavelengths of light and sound waves, respectively, and *L* is a length of AO interaction), the diffraction pattern consists of two maxima (the zeroth and the first orders). The first diffraction order summarizes selective reflections of light from the wave fronts of ultrasonic waves and thus is informative for the analysis of the light spectrum.

Being compact, solid-state and PC-controlled, AO tunable filters (AOTFs) are a robust and versatile tool to manage the spectrum of transmitted light [17,18]. Arbitrary spectral access, multi-wavelength selection and transmission function apodization are key features of AO interaction that deliver a good compromise in terms of spectral bandwidth, tuning range, aperture and switching time.

Thus, Bragg diffraction of white light using multiple ultrasound waves seems an attractive basis for building reliable and flexible CRSs. In this study, we experimentally demonstrate the feasibility of this technique for accurate color reproduction.

## 2. Proposed Approach

The concept of the proposed AOTF-based CRS is shown in Figure 2. It includes an AO crystal and a piezoelectric transducer bonded to it. When electrical signals are applied to the transducer, it generates ultrasonic waves propagating through the crystal. These waves modulate the refractive index and thereby form a three-dimensional diffraction grating. The simplest grating is achieved when applying a single acoustic frequency *f*_0_. In this mode with the Bragg phase-matching conditions fulfilled, there is only the first diffraction order, and the AOTF transmission function is described with a squared sinc function: *T*(*λ*) ~ sinc^2^((*λ* − *λ*_0_)/*δλ*), where *λ*_0_ ~ 1/*f*_0_ and *δλ* ~ *λ*_0_^2^ [16]. Diffraction light intensity is proportional to the acoustic power *P* radiated from the transducer. If *N* frequencies *f_j_* (*j* = 1, 2 … *N*) are launched simultaneously, then *T*(*λ*) is the sum of transmission functions *T*(*λ* − *λ_j_*) defined by each of them. By varying the values of *N*, *f_j_* and *P_j_*, one can select the required number *N* of the spectral bands as well as precisely tune their positions *λ_j_* and intensities *I_j_*.

The number of selected bands *N* determines the palette of available colors. If the AOTF operates in dual-frequency mode, then reproducible colors lay on the line segment between the points defined by *λ*_1_ and *λ*_2_ (Figure 3a). Changing the frequencies *f*_1_ and *f*_2_ varies the inclination angle of this line on the CIE XYZ 1931 chromaticity plane and thus achieves segments with different colors. The proper power ratio *P*_1_/*P*_2_, i.e., intensities *I*_1_/*I*_2_, enables accurate selection of a particular color within the segment. If the AOTF selects 3, 4 or more spectral bands simultaneously, then the available gamut is described using a triangle (Figure 3b), quadrangle (Figure 3c) or polygon with more vertices defined by the selected wavelengths *λ_j_*, i.e., the acoustic frequencies *f_j_* launched in the crystal. The relative intensities *I_j_* of the light beams in this case have to be assigned in accordance with the center of gravity law [19] and, therefore, the acoustic power *P_j_* has to be adjusted accordingly.

Thus, to reproduce the color *x*, *y* with an AOTF-based CRS, it is enough in principle to choose the number *N* and positions *λ_j_* of the basic wavelengths, calculate the relative intensities *I_j_* of the diffracted beams and properly adjust the values of acoustic power *P_j_*.

## 3. Experimental Setup

Birefringent crystals are the most efficient AO medium in terms of diffraction efficiency, availability and ease of handling. They enable anisotropic AO interaction, which changes the polarization (e→o or o→e) of the incident optical beam and enables easy separation of the first diffraction order from the zeroth one using a linear polarizer [16]. Tellurium dioxide (TeO_2_) is one of the best AO crystals that is widely used because of its high figure of merit *M*_2_ = *n*_o_^3^
*n*_e_^3^*p*^2^/*ρV*^3^ (*n*_o_ and *n*_e_ are refractive indices for ordinary and extraordinary polarized light, *p* is effective photoelastic constant, *ρ* is density and *V* is ultrasound wave velocity), which is 1200 × 10^−15^ s^3^kg^−1^ at the wavelength *λ* = 632.8 nm. A high value of *M*_2_ ensures the low driving power *P =* (*λ*^2^*h*)/(2*M*_2_*l*) (*h* and *l* are the height and length of the acoustic beam, respectively) that is required for the high efficiency of Bragg diffraction and eases the fabrication of AO devices.

For our experiments, we produced a typical TeO_2_ AOTF with a cut angle of 7° and a wide-aperture geometry of AO interaction. The AOTF had an optical aperture of 10 × 10 mm and angular aperture of 4° × 4°. The front facet of the crystal was perpendicular to the incident light beam. For easier mixing of the diffracted light beams, the back facet angle of the crystal corresponds to their minimal chromatic shift in the range of 430–780 nm [20]. Because of the poor transparency of tellurium dioxide below 430 nm, the violet part 380–430 nm of the visible spectrum is unavailable. With some other AO crystals, e.g., quartz (SiO_2_), it is possible to cover the entire visible range of 380–780 nm.

To generate bulk ultrasonic waves in the TeO_2_ crystal, we bonded a thin-plate piezoelectric transducer to it via cold indium welding. The transducer had 2 3.5 × 8 mm sections with a 0.5 mm gap between them. The sections were electrically connected in series to reduce the effective capacitance. The thickness of the transducer corresponds to the acoustic frequency 85 MHz which is the central frequency of the operating range. The transducer was made of *X*-cut lithium niobate (LiNbO_3_) because of its large piezoelectric and electromechanical coupling coefficients, stable functioning at high frequencies and well-established manufacturing process. To place ground and drive electrodes, thin films of chromium and gold were evaporated and deposited on the transducer. To ensure a good acoustic impedance matching of the piezoelectric transducer and AO crystal, a specific acoustic bond consisting of chromium, gold and indium was evaporated. After welding, electrical matching networks were installed to match the transducer and the electrical impedance of the driving radio frequency generator. When a radio frequency signal was applied to the piezoelectric transducer, it vibrated and generated bulk shear ultrasonic waves that propagated in the TeO_2_ crystal and were absorbed by the acoustic absorber placed at its opposite facet.

By varying the acoustic frequency *f* from 56 to 137 MHz, the wavelength *λ* of filtered light may be precisely tuned in the range 430–850 nm according to the tuning curve *λ*(*f*) obtained after AOTF calibration using a certified monochromator. The total driving acoustic power *P* = Σ*P_j_* is variable from 0 to 2.5 W and defaults to 1 W at all wavelengths. To enable multi-frequency mode, we used a four-channel driver based on a direct digital synthesizer (Analog Devices AD9959) and two homemade two-channel amplifiers, which enabled precise assignment of the position and intensity of each selected spectral band. 

Figure 4 illustrates the experimental setup. To ensure anisotropic o→e diffraction and cut off non-diffracted light, we placed the AO cell between crossed polarizers P1 and P2. Lens L1 with a focal distance of 35 mm collimated wide-band light from a fiber-coupled 150 W halogen lamp LS and sent it to the AOTF. After AO diffraction, lens L2 with a focal distance of 50 mm focused the filtered light onto the sensor of color camera CAM (TheImagingSource DFM 42BUC03). CAM acquired images of the diffracted beams and thus facilitated the comparison of the reproduced colors with the reference ones. Its quantum efficiency is spectrally dependent and must be taken into account during color rendering experiments. That is why, between L2 and CAM, there was a 50/50 beam splitter BS that directed the reflected light to spectrometer SP (Ocean Insight Flame-T-UV-VIS).

SP is necessary for the interactive measurement and correction of the light intensities *I_j_* in the selected bands and managing the acoustic power *P_j_* in AOTF channels with respect to the quantum efficiency of the CAM sensor in order to ensure that the ratio of the intensities corresponds to the assigned color.

All the experiments presented below were carried out in typical laboratory conditions (temperature 23 °C and relative humidity 50%). In fact, all optical (lenses, polarizers, AOTF, etc.) and electronic (LS, AOTF driver, CAM) components of the setup can operate at a wider range of ambient temperatures (10–40 °C) and humidity (30–70%). To validate the repeatability of the proposed CRS in various tolerable combinations of temperature and humidity, further study is necessary.

## 4. Experimental Results

We demonstrated the proposed system’s capabilities by conducting multiple color reproduction experiments. Figure 5 shows images of the reference and reproduced colors obtained with an AOTF operating in single- (points A, E and I), dual- (points B, D and G) and three-frequency (points C, F and H) modes. Here, the amount and positions of the colors for color rendering as well as the spectral bands for their reproduction were selected just to illustrate the flexibility of the AOTF-based CRS. Depending on the application, these parameters may be optimized for the stable and repeatable reproduction of colors located in a certain area of the CIE XYZ 1931 chromaticity plane without spectral tuning. The images formed by diffracted light beams are presented after correction, i.e., averaging over the cross-section of the diffracted beams’ profiles. Otherwise, they look non-uniform because of vignetting, diffraction-caused distortions and optical aberrations (see below in Figure 6d). The exposure time was equal to 20 ms in all experiments, as shown in Figure 6. The relative intensities presented in Figure 6 were calculated from the measured values with respect to the quantum efficiency of the camera sensor. 

Figure 5 demonstrates the high visual similarity of the assigned colors and the colors in the acquired images. To obtain a more objective merit of color rendition, we took a certified color checker (Edmund Optics Rez Checker) widely used for testing true color balance and comparing with reference patterns. Figure 6a shows the color checker’s image acquired with the same lens L2 and the same color camera CAM operating in the same mode as in the color reproduction experiments [18] (Figure 6b). We chose the wavelengths *λ*_1_ = 460 nm, *λ*_2_ = 520 nm and *λ*_3_ = 650 nm as the basic ones to form a wide enough triangle in CIE XYZ 1931 and cover all 19 selected colors (Figure 6c). In this experiment, we again calculated the necessary ratio of the intensities *I_j_* and applied the acoustic power *P_j_* accordingly. The total acoustic power Σ*P_j_* applied to the AOTF was adjusted in order to achieve the same intensity of the maximal component in all acquired images as in the reference ones shown in Figure 6b. Figure 6d,e illustrates the raw and corrected (averaged) color images formed by the diffracted light. Visually, the reproduced colors (Figure 6e) seem almost indistinguishable from the ones in the color checker (Figure 6a). Correspondingly, the average and median reproduction quality according to Δ*E*_2000_ are about 1.25 and 1.12, respectively, which is close to the just noticeable difference (Δ*E*_2000_ = 1) threshold, while those for the angular errors between the reference and generated tristimulus are about 0.68 and 0.61, respectively.

Figure 7 illustrates the spectra *I*(*λ*) selected using AOTF and measured with spectrometer SP for all 19 colors in the color checker. Coordinates *x*, *y* of each color as well as the calculated relative intensities *I*_1_:*I*_2_:*I*_3_ of the respective monochromatic components 460 nm, 520 nm and 650 nm necessary for its reproduction are shown inside the circles.

## 5. Discussion

Though the CRS implementation demonstrated in this study has multiple advantages including almost full CIE XYZ space coverage, high speed and repeatability, it still has the potential for higher efficacy. 

First, accurate spatio-temporal and radiometric calibration of the setup will allow obtaining the dependence of image intensity *I* on acoustic power *P* in each image pixel with respect to the transmittance of optical components and the quantum efficiency of the sensor and thus will simplify the system by excluding the spectrometer. Regular calibration and refinement of correction factors will compensate for long-term temporal variations in CRS performance.

Second, accurate optical CRS design will lead to uniform and non-distorted light beams acceptable for human visual color perception study in the entire color triangle. To achieve this, the simulation and optical design of the setup must take into account multiple effects related to light transmittance through fiber-optic cable and AOTFs: material and waveguide dispersion, scattering and absorption losses, wave-front distortions caused by AO diffraction, etc.

Third, the adjustable width *δλ* and shape *T*(*λ*) of the transmission function, which is an important advantage of AO diffraction, was not used in this feasibility study but is quite helpful in many rendering tasks and may widen the variety of AOTF-based CRS applications. A common approach to interactive management of the transmission function is dispersive synthesis, i.e., varying the ultrasonic signal with a complex-valued spectrum [17]. With this feature, the proposed CRS may become even more flexible and versatile and may deliver multiple scenarios of color reproduction. Interactive and broad-range change in the number *N*, shapes *T_j_*(*λ*) and intensities *I_j_*(*λ*) of the selected spectral channels is an ultimate solution for multiple color rendering tasks but requires advanced hardware for programmable radio frequency waveform synthesis.

Fourth, one of the main AOTF drawbacks critical for some color reproduction applications is its low optical throughput due to its small angular aperture, narrow spectral band, linear polarization selection under anisotropic diffraction and other factors. Depending on the application, this problem may be solved in a few ways: using a light source with higher power, optimizing the optical coupling between the AOTF and other components, frequency modulation of ultrasonic waves [21], complicating the optical scheme to ensure simultaneous effective AO diffraction of both polarization components of a randomly polarized light, etc.

Fifth, the metrological characteristics of the proposed AOTF-based illuminants must be further investigated. Evaluation of their accuracy and repeatability is a challenging task and requires detailed theoretical and experimental analysis of the contribution of multiple systematic and random factors to the color reproduction error. Spatial, temporal, spectral and intensity variations are inevitable in systems of this type and must be calibrated over the entire range of multiple physical parameters with respect to the operating mode and conditions.

With these issues taken into account, the proposed AOTF-based approach to color rendering may have multiple potential applications in biomedicine, industry and metrology. One of them deserves more thorough consideration. Now, there are still no color-appearance models that ensure high-quality color representation and prediction within the entire color triangle of a standard observer. One of the main reasons for this is the lack of tools and techniques for studying the color perception of an arbitrary reference color located close to the border of the color triangle. The most popular methods are based on wide-coverage displays and do not allow such a capability. Colorimetric setups require installation and changing multiple certified color filters, which makes large-scale measurements almost impossible. The new AOTF-based technique proposed in this study may help overcome these limitations and bring new opportunities to vision science. It demonstrates an efficient method of building laboratory setups for human vision system research including presenting stimuli, measuring thresholds and quantitative characteristics and analyzing the color perception properties. Such systems will be a pioneer in understanding human visual perception over the entire area of the color triangle of the standard CIE XYZ 1931 observer.

## 6. Conclusions

AO interaction is a flexible tool for the spectral filtration of light by varying the parameters of the acoustic waves. AO interaction enables the stable selection of monochromatic components with the required wavelengths and amplitudes. Launching a few ultrasound frequencies with tunable power in AO crystal and mixing the diffracted light beams can enable the creation of an almost infinite number of colors. Together with comprehensive radiometric calibration of the optical setup and digital image processing, this technique delivers unique color reproduction capabilities with just one white light source. This type of tunable illuminant provides nearly full coverage of the CIE XYZ 1931 chromaticity space and demonstrates high color-rendering quality, high luminous efficiency, fast tuning and easy adjustment. We believe that an AOTF-based CRS may have multiple applications in industry and biomedicine and can be helpful for developing new up-to-date color reproduction standards in colorimetry that are extremely important for the worldwide scientific community.

## 7. Patents

Machikhin A., Beliaeva A., Romanova G.: color reproduction method based on polychromatic acousto-optic filtering of broadband radiation 17 March 2022 (RU 2786365 C1) (In Russian).

## Figures and Tables

**Figure 1 materials-16-04382-f001:**
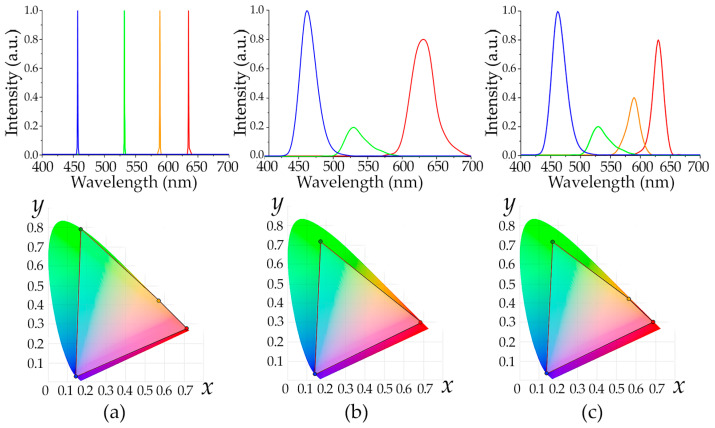
Typical SPDs (upper row) of CRSs based on four lasers (**a**), three (**b**) and four LEDs (**c**) and color areas that they cover in CIE XYZ 1931 chromaticity space (lower row).

**Figure 2 materials-16-04382-f002:**
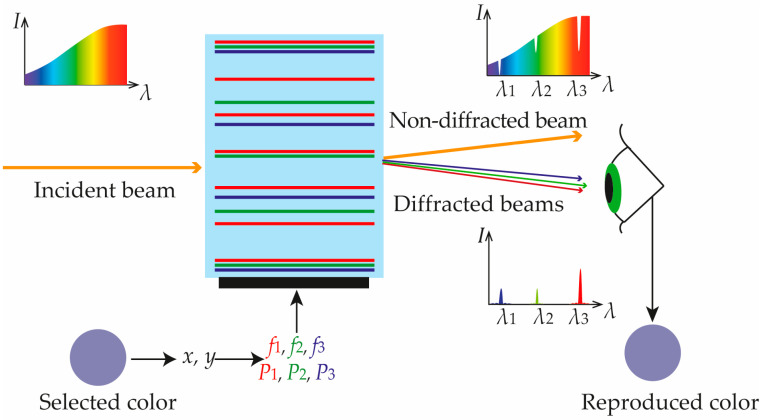
Proposed CRS concept.

**Figure 3 materials-16-04382-f003:**
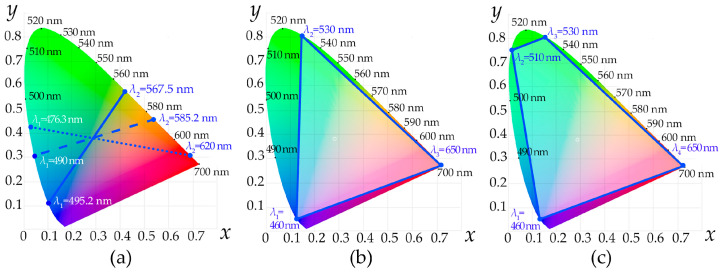
Сolor areas covered in CIE XYZ 1931 chromaticity space with AOTF-based CRS operating in dual- (**a**), three- (**b**) and four-frequency (**с**) mode.

**Figure 4 materials-16-04382-f004:**
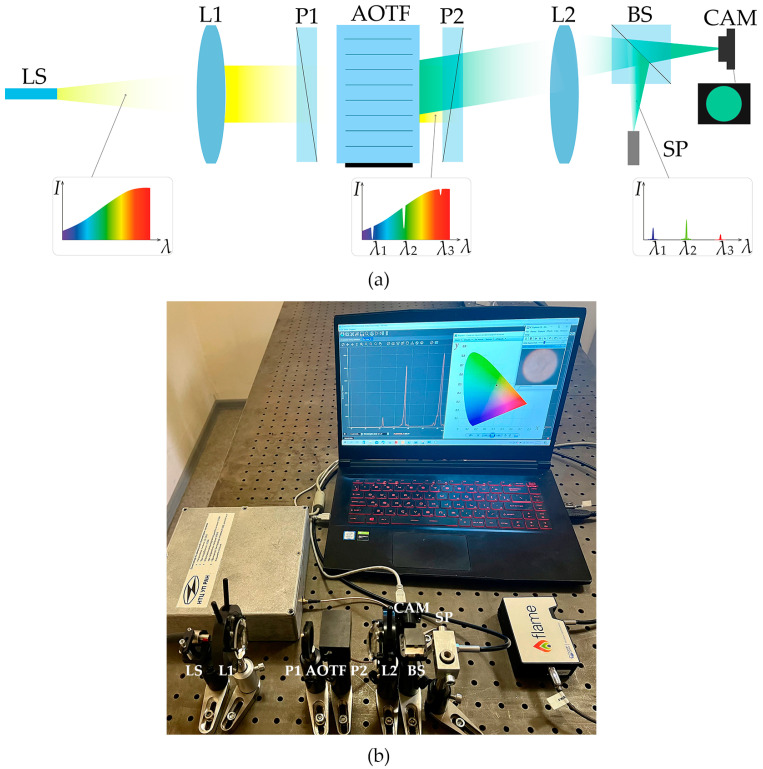
Scheme (**a**) and appearance (**b**) of the experimental setup for color reproduction. LS—light source, L1 and L2—lenses, P1 and P2—polarizers, AOTF—acousto-optical tunable filter, BS—beam splitter, CAM—color camera, SP—spectrometer.

**Figure 5 materials-16-04382-f005:**
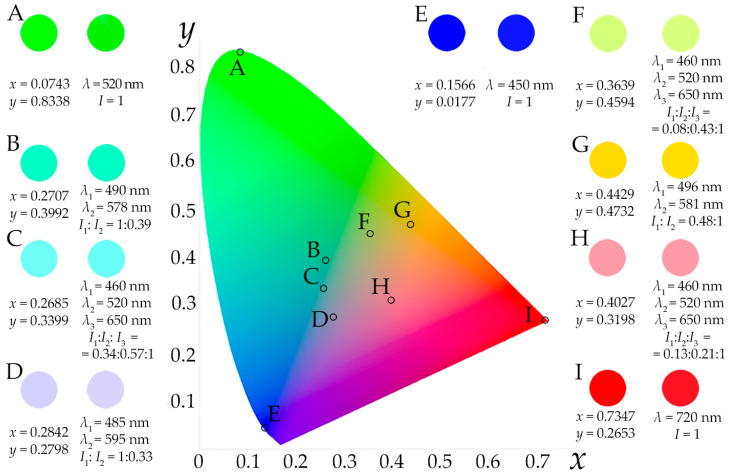
Reference (left circles) and reproduced (right circles) colors plotted in CIE XYZ 1931 space. Reference colors are assigned according to their *x*, *y* coordinates. Reproduced colors are presented according to the wavelengths *λ_j_* and relative intensities *I_j_* of the selected monochromatic components.

**Figure 6 materials-16-04382-f006:**
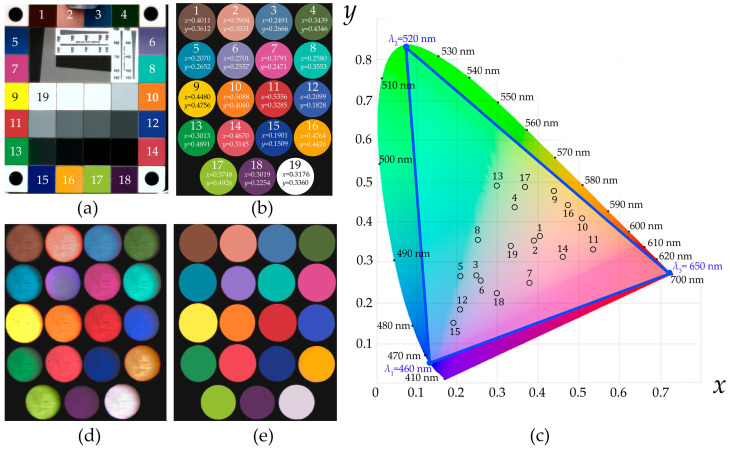
Color checker (**a**), its selected colors presented as reference images (**b**) and plotted in CIE XYZ 1931 space (**c**) and acquired raw (**d**) and averaged (**e**) images.

**Figure 7 materials-16-04382-f007:**
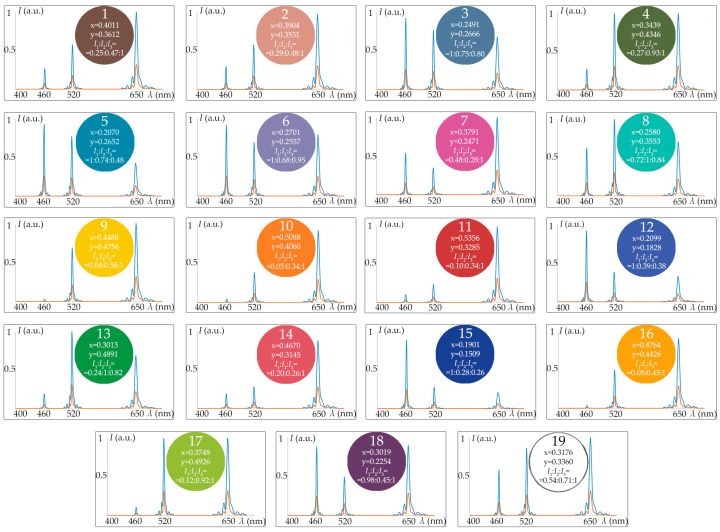
Spectra required to reproduce 19 colors in color checker for visual perception (red) and digital registration (blue), i.e., regardless of and with regard to the spectral quantum efficiency of the camera sensor, respectively.

## Data Availability

The data that support the findings of this study are available from the corresponding author upon reasonable request.

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
