# Peer review of "Color Reproduction by Multi-Wavelength Bragg Diffraction of White Light"

_materials, 2023, doi:10.3390/ma16124382_

Round 1
Reviewer 1 Report
This paper proposes a method to reproduce color based on multi-bandpass acousto-optic (AO) filtration of white light. The authors also verify this method with multiple experiments.
Its mechanism is easy to understand. The authors present the experiments clearly. In the end, the authors mention their patents utilizing this technique. That is a good work.
This technique has the great value in the engineering field. I highly suggest that the author can write more about the scenarios how this technique can be applied. If there is some preliminary research done by others, I also suggest the authors can write some.
About the table and figure, I have a question about Fig 6(a). I notice that there is a table demonstrating “dots per inch”. May I know whether it is relevant with this article? If it is, can the authors explain what it is for?
In summary, I think this article can be accepted after some improvements.
Author Response
Thank you for kind assessment, patience, thorough analysis and valuable comments. To address your remarks, we have revised the text. All changes are marked with green color.
- I highly suggest that the author can write more about the scenarios how this technique can be applied. If there is some preliminary research done by others, I also suggest the authors can write some.
We expanded Discussion by the description of potential scenarios and applications of the proposed technique.
- About the table and figure, I have a question about Fig 6(a). I notice that there is a table demonstrating “dots per inch”. May I know whether it is relevant with this article? If it is, can the authors explain what it is for?
This is the image of a standard color checker (https://www.edmundoptics.com/p/rez-checker-matte/29795/). It includes not only color patches but also a resolution test target that indeed was not used in this study.
Reviewer 2 Report
The submitted manuscript is almost excellent. Clear, concise, balanced, nicely illustrated and presents a piece of a good science with practical emphasis. My suggestions are minor ones:
1)Lines 87-89, is there a software that allows to do those calculations?
2)Line 91, it should be “TeO2”
3)Line 93, it should be “mm”
4)Line 94, what about the 380-450 nm range? Is it still covered?
5)Line 112, the symbols used in this figure must be explained in the Figure caption
6)Figure 7, how can you describe the differences between the observed and predicted intensities? Figure 7, it is clear that the wavelengths have been reproduced correctly. But what about the relative intensities of the signals? It is difficult to compare when the spectra are not normalized, i.e. that the highest signals have the same intensity.
7)Line 165, it should be “.” before “First”, not “,”.
The Authors state that they have used the piezoelectric transducer. However, information about this device is missing such as strain sensitivity, threshold, span to threshold ratio, principle of operation. Please update this data in the Materials&Method section. Also, the manufacturer of this part should be stated.
Figure 5, what was the reason behind choosing those particular colors (points)? I appreciate that the Authors have used the “edge” points (A, E, I) but most of the points are located in the middle of the scale. Therefore, there are large color areas that haven’t been sampled at all,i.e. low x and large y.
It is not stated in the manuscript if the temperature and humidity have been maintained at constant values or even have they been monitored? Also, the ranges of working conditions (humidity, temperature) is not stated. I understand that the Authors haven’t done such experiments to check this but they should at least describe the possible limitations of this methods in terms of external conditions.
English is OK
Author Response
Thank you for kind assessment and valuable comments. To address your remarks, we have revised the text. All changes are marked with green color.
- Lines 87-89, is there a software that allows to do those calculations?
This our own software. It allows AOTF calibration (frequency and power) and interactive control as well as simple calculation of the wavelengths and ratio of intensities necessary reproduce a color with specific CIE XYZ 1931 coordinates.
- Line 91, it should be “TeO2”
Corrected.
- Line 93, it should be “mm”
Corrected.
- Line 94, what about the 380-450 nm range? Is it still covered?
This range is unavailable due to poor transparency of TeO2 crystal below 430 nm. With other crystals, e.g. quartz, the proposed approach can cover the entire visible range (380-780 nm).
- Line 112, the symbols used in this figure must be explained in the Figure caption
Corrected.
- Figure 7, how can you describe the differences between the observed and predicted intensities? Figure 7, it is clear that the wavelengths have been reproduced correctly. But what about the relative intensities of the signals? It is difficult to compare when the spectra are not normalized, i.e. that the highest signals have the same intensity.
We added predicted relative intensities of monochromatic components necessary for color reproduction in Fig. 7 so that the reader can compare them with the measured spectra.
- Line 165, it should be “.” before “First”, not “,”.
Corrected.
- The Authors state that they have used the piezoelectric transducer. However, information about this device is missing such as strain sensitivity, threshold, span to threshold ratio, principle of operation. Please update this data in the Materials&Method section. Also, the manufacturer of this part should be stated.
We added a detailed description of the piezoelectric transducer.
- Figure 5, what was the reason behind choosing those particular colors (points)? I appreciate that the Authors have used the “edge” points (A, E, I) but most of the points are located in the middle of the scale. Therefore, there are large color areas that haven’t been sampled at all,i.e. low x and large y.
The idea of this Figure was to demonstrate color reproduction with AOTF operating in single- (points A, E and I), dual- (points B, D and G) and three-frequency (points C, F and H) modes. In fact, it is not a problem to reproduce almost any points. Figure 6 shows a wider scope of reproduced colors.
- It is not stated in the manuscript if the temperature and humidity have been maintained at constant values or even have they been monitored? Also, the ranges of working conditions (humidity, temperature) is not stated. I understand that the Authors haven’t done such experiments to check this but they should at least describe the possible limitations of this methods in terms of external conditions.
During the experiments, we maintained conventional laboratory conditions, i.e. temperature 23°C and relative humidity 50%. We added these numbers in the manuscript. In fact, all components of our setup, i.e. light source, AOTF and camera, are quite robust and can operate in a wide range of outer temperature (10-40°C) and humidity (30-70%). To validate this, further study is necessary.
Reviewer 3 Report
Authors showed reproduction of multi-wavelength of white light. The page of the article is show. Author might change the manuscript type to communication. English grammar looks fine. There are some comments which need to be udpated below.
1. Please use abbreviated journal name in Reference section. Please see author guidelines of MDPI.
2. Please add ref. (By adjusting relative intensities of ~) with ref. (https://www.mdpi.com/1424-8220/18/10/3324)
3. Please show specification of a piezoelectric transducer as described in Line 59. such as frequency and apeture.
4. Please use formal English expression. In Line 87, author had better change so to therefore or thus.
5. Please show manufacturer information of the transducer in Line 92.
6. Please change Fig. to Figure in Line 100.
7. Figure 7 has small label and font sizes.
8. Authors had better discuss the limitation and future work in conclusion or discussion sections.
9. Please describe CIE XYZ 1931 in detail.
10. Please provide city and country information in conference paper.
None
Author Response
Thank you for your valuable comments. To address your remarks, we have revised the text. All changes are marked with green color.
- Please use abbreviated journal name in Reference section. Please see author guidelines of MDPI.
Corrected.
- Please add ref. (By adjusting relative intensities of ~) with ref. (https://www.mdpi.com/1424-8220/18/10/3324)
We added this paper to the reference list.
- Please show specification of a piezoelectric transducer as described in Line 59. such as frequency and aperture.
We added a detailed description of the piezoelectric transducer.
- Please use formal English expression. In Line 87, author had better change so to therefore or thus.
Corrected.
- Please show manufacturer information of the transducer in Line 92.
Actually, the transducer is home-made. We are buying just LiNbO3 crystals from a few suppliers and bond them ourselves to TeO2 cells via cold indium welding.
- Please change Fig. to Figure in Line 100.
Corrected.
- Figure 7 has small label and font sizes.
We increased label and font sizes in Fig. 7.
- Authors had better discuss the limitation and future work in conclusion or discussion sections.
We expanded Discussion by the description of the ways to improve the efficiency of the proposed technique and its potential applications.
- Please describe CIE XYZ 1931 in detail.
We added detailed description of CIE XYZ 1931.
- Please provide city and country information in conference paper.
Corrected.